# Optimal Transport for Long-Tailed Recognition with Learnable Cost Matrix

**Hanyu Peng, Mingming Sun, Ping, Li**

Cognitive Computing Lab
Baidu Research
No.10 Xibeiwang East Road, Beijing 100193, China
10900 NE 8th St. Bellevue, Washington 98004, USA
{penghanyu,sunmingming01,liping11}@baidu.com

## Abstract

It is attracting attention to the long-tailed recognition problem, a burning issue that has become very popular recently. Distinctive from conventional recognition is that it posits that the allocation of the training set is supremely distorted. Predictably, it will pose challenges to the generalisation behaviour of the model. Approaches to these challenges revolve into two groups: firstly, training-aware methods, with the aim of enhancing the generalisability of the model by exploiting its potential in the training period; and secondly, post-hoc correction, liberally coupled with training-aware methods, which is intended to refine the predictions to the extent possible in the post-processing stage, offering the advantages of simplicity and effectiveness. This paper introduces an alternative direction to do the post-hoc correction, which goes beyond the statistical methods. Mathematically, we approach this issue from the perspective of optimal transport (OT), yet, choosing the exact cost matrix when applying OT is challenging and requires expert knowledge of various tasks. To overcome this limitation, we propose to employ linear mapping to learn the cost matrix without necessary configurations adaptively. Testing our methods in practice, along with high efficiency and excellent performance, our method surpasses all previous methods and has the best performance to date.

## 1 Introduction

Classification problems in the real world are generally challenged by the long-tailed label distribution, i.e., having a small number of samples for a majority of labels, and a dominant number of samples for a minority of labels (Van Horn & Perona, 2017; Buda et al., 2018; Liu et al., 2019). It is also known as imbalanced recognition, which has been widely studied in the past decades (Cardie & Nowe, 1997; Chawla et al., 2002; Qiao & Liu, 2009; Cui et al., 2019). These distribution biases pose a significant challenge to predictive modeling; conceivably, models often suffer from poor generalisation and undesirable estimation bias (Cao et al., 2019; Kang et al., 2020; Zhou et al., 2020).

Recently, a renewed interest in the problem of long-tail recognition has emerged following the context of neural networks, as numerous publications in the literature endeavour to resolve the problem albeit in different ways including decouple (Kang et al., 2020), meta-learning (Ren et al., 2020; Wang et al., 2020; Li et al., 2021), post-hoc correction (Tang et al., 2020; Hong et al., 2021), etc (Liu et al., 2019; Cao et al., 2019; Tang et al., 2020). One of the representative methods of post-hoc correction, Logit Adjustment Menon et al. (2021), provides a statistical correction to the prediction, receiving widespread attention for its simplicity and validity. But the downside is that it is conducted on individual samples, the rectified marginal distribution may not satisfy the desired distribution.

Figuring out exact flaws of Logit Adjustment, our explicit modeling of the problem mathematically turns into an equational constraint, meanwhile to minimise the difference between refined distribution and the original one, this minimisation is motivated upon the inner-product similarity. A little further, the resulting problem can be linked to OT. Drawing on this linkage, we develop it further by proposing

a linear mapping to automatically learn cost matrix, thereby circumventing the requirement for expert knowledge to configure this matrix. In summary, our **contributions** are as follows:

- We propose an alternative direction based on convex optimisation to do post-hoc correction, which goes beyond previous direction from the statistical view.

- Imposing marginal distributions to align ideal ones, we derive an optimisation problem tied to OT that is solved using Sinkhorn. More further, for better learning of the cost matrix, we present a linear mapping enabling elegant learning with one-layer network.

- The experimental evidence shows the high efficiency and best performance on three benchmarks. It verifies that addressing the post-hoc problem via OT is helpful and effective.

## 2 PRELIMINARIES

In this section, we begin with notational definition, followed by an introduction to the long-tailed recognition problem. Finally, we briefly review the OT and Logit Adjustment Menon et al. (2021).

**Notations**: In what follows, for two matrices $\boldsymbol{X}, \boldsymbol{Y} \in \mathbb{R}^{N \times K}$, we denote $\langle \boldsymbol{X}, \boldsymbol{Y} \rangle = \sum_{n=1}^{N} \sum_{k=1}^{K} \boldsymbol{X}_{nk} \boldsymbol{Y}_{nk}$ as the Frobenius dot-product. $\delta(\cdot)$ stands for the Dirac function, $p(\cdot)$ represents the probability distribution. $U(\boldsymbol{r}, \boldsymbol{c}) = \{\boldsymbol{P} \in \mathbb{R}_{+}^{N \times K} | \boldsymbol{P} \mathbf{1}_K = \boldsymbol{r}, \boldsymbol{P}^{\mathsf{T}} \mathbf{1}_N = \boldsymbol{c}\}$, where $\mathbf{1}_N$ and $\mathbf{1}_K$ are $N$-dimension and $K$-dimension vector whose elements are all 1. $\boldsymbol{r}$ and $\boldsymbol{c}$ refer to the vectors of size $N$ and $K$, $U(\boldsymbol{r}, \boldsymbol{c})$ include all matrices with row and column sums $\boldsymbol{r}$ and $\boldsymbol{c}$ respectively.

### 2.1 PROBLEM FORMULATION

Having a collection of training samples $\{(\boldsymbol{x}_n^s, y_n^s)\}_{n=1}^{N_s}$, validation samples $\{(\boldsymbol{x}_n^v, y_n^v)\}_{n=1}^{N_v}$ and test samples $\{(\boldsymbol{x}_n^t, y_n)^t\}_{n=1}^{N_t}$ for classification with $K$ labels and input $\boldsymbol{x} \in \mathbb{R}^d$, long-tailed recognition assumes that the class-prior distribution for training data $p(y^s)$ is different from that for validation data $p(y^v)$ and test data $p(y^t)$. Specifically, *long-tailed recognition means the distribution $p(y^s)$ is highly skewed, that is, some classes have the dominant number of samples, while tailed labels own a very small number of samples*. We can use imbalance ratio to measure the skewness in training data set, which can be defined as $R = \frac{N_{max}^s}{N_{min}^s}$, where $N_{max}^s$ and $N_{min}^s$ denote the largest and smallest number of samples in the training data set, respectively. *In this paper, we assume that the marginal distribution of the test set is known, we consider it as an implicit prior knowledge to be applied.* Stepping back, even if we do not know the marginal distribution of the test dataset in advance. There are still ways to estimate the marginal distribution of the test dataset relatively precisely, such as methods in Hendrycks et al. (2018); Azizzadenesheli et al. (2019).

Obviously, most models trained on imbalanced training data set would suffer from extremely limited generalisation ability. Hence the ultimate goal is to learn a model that minimises the empirical risk:

$$\mathcal{J}\left(\Phi\left(\boldsymbol{x}_n^s\right), y_n^s\right) = \frac{1}{N_s} \sum_{n=1}^{N_s} \mathcal{L}\left(\Phi(\boldsymbol{x}_n^s), y_n^s\right), \tag{1}$$

where $\Phi(\boldsymbol{x}_n^s) \in \mathbb{R}^K$ denotes logits with associated sample, $\Phi(\cdot) : \mathbb{R}^d \to \mathbb{R}^K$ represents the mapping via neural networks, $\mathcal{L}$ stands for the loss function, typically cross entropy for classification problem.

### 2.2 REMINDERS ON OPTIMAL TRANSPORT

OT is used to calculate the cost of transporting one probability measure to another. We next present a brief introduction to OT to help us better view the long-tailed problem from an OT perspective.

For two random variables $X$ and $Y$, we denote its corresponding probability measures as $r$ and $c$. Besides, $\mathcal{C}(X, Y) : X \times Y \to \mathbb{R}_+$ stands for cost function which measures the expense of transporting $X$ to $Y$. Based on these, we can define OT distance between $X$ and $Y$ as

$$d(r, c) = \min_{\boldsymbol{\pi} \in \Pi(r,c)} \int_{X \times Y} \mathcal{C}(\boldsymbol{x}, \boldsymbol{y}) \pi(\boldsymbol{x}, \boldsymbol{y}) d\boldsymbol{x} d\boldsymbol{y}, \tag{2}$$

where $\Pi(r,c) = \left\{ \int_{\mathcal{Y}} \pi(\boldsymbol{x},\boldsymbol{y})d\boldsymbol{y} = r, \int_X \pi(\boldsymbol{x},\boldsymbol{y})d\boldsymbol{x} = c \right\}$ is the joint probability measure with $r$ and $c$. When we extend the above to the discrete situation, we consider following discrete distributions:

$$\boldsymbol{r} = \sum_{i=1}^{N} p_i(\boldsymbol{x}_i)\delta(\boldsymbol{x}_i) \quad \boldsymbol{c} = \sum_{j=1}^{K} p_i(\boldsymbol{y}_j)\delta(\boldsymbol{y}_j) \tag{3}$$

where $p_i(\boldsymbol{x}_i)$ and $p_i(\boldsymbol{y}_j)$ represent the probability mass to the sample $\boldsymbol{x}_i$ and $\boldsymbol{y}_j$ respectively. In this context, OT distance can be expressed as:

$$d_{\boldsymbol{M}}(\boldsymbol{r},\boldsymbol{c}) = \min_{\boldsymbol{P}\in U(\boldsymbol{r},\boldsymbol{c})} \langle \boldsymbol{P},\boldsymbol{M}\rangle. \tag{4}$$

where $\boldsymbol{M}$ stands for the cost matrix constructed by $\boldsymbol{M}_{ij} = \mathcal{C}(\boldsymbol{x}_i,\boldsymbol{y}_j)$. The goal of OT is to find a transportation matrix $\boldsymbol{P}$ that minimizes the distance $d_{\boldsymbol{M}}(\boldsymbol{r},\boldsymbol{c})$

As we can see, OT is a distance measure between two probability distributions under some cost matrix (Villani, 2008). However, when we use network simplex or interior point methods to solve the above optimisation problem, it often comes at the cost of heavy computational demands. To tackle this issue, OT with entropy constraint is proposed to allow the optimisation at small computational cost in sufficient smoothness (Burges et al., 2013). By adding a Lagrangian multiplier to the entropy constraint, the new formulation can be defined as follows:

$$d_{\boldsymbol{M}}^{\lambda}(\boldsymbol{r},\boldsymbol{c}) = \langle \boldsymbol{P}^{\lambda},\boldsymbol{M}\rangle \qquad \text{where} \quad \boldsymbol{P}^{\lambda} = \operatorname*{arg\,min}_{\boldsymbol{P}\in U(\boldsymbol{r},\boldsymbol{c})} \langle \boldsymbol{P},\boldsymbol{M}\rangle - \lambda h(\boldsymbol{P}), \tag{5}$$

where $\lambda \in [0,+\infty]$, $h(\boldsymbol{P}) = -\sum_{n=1}^{N}\sum_{k=1}^{K} \boldsymbol{P}_{nk}\log \boldsymbol{P}_{nk}$, $d_{\boldsymbol{M}}^{\lambda}(\boldsymbol{r},\boldsymbol{c})$ is also known as dual-Sinkhorn divergence, besides, it can be calculated with matrix scaling algorithms for cheaper computational demand. The following lemma guarantees the convergence and uniqueness of the solution.

**Lemma 1** *For $\lambda > 0$, the solution $\boldsymbol{P}^{\lambda}$ is unique and has the form $\boldsymbol{P}^{\lambda} = \boldsymbol{diag}(\boldsymbol{u})\boldsymbol{K}\boldsymbol{diag}(\boldsymbol{v})$, where $\boldsymbol{u}$ and $\boldsymbol{v}$ are two non-negative vectors uniquely defined up to a multiplicative factor and $\boldsymbol{K} = e^{-\boldsymbol{M}/\lambda}$ is the element-wise exponential of $-\boldsymbol{M}/\lambda$.*

The above lemma states the uniqueness of $\boldsymbol{P}^{\lambda}$ (Sinkhorn, 1974), and $\boldsymbol{P}^{\lambda}$ can be efficiently computed via Sinkhorn's fixed point iteration $\boldsymbol{u},\boldsymbol{v} \leftarrow \boldsymbol{r}./\boldsymbol{K}\boldsymbol{v}, \boldsymbol{c}./\boldsymbol{K}^{\intercal}\boldsymbol{u}$.

### 2.3 A QUICK RECAP OF LOGIT ADJUSTMENT

We give a brief introduction to Logit Adjustment (Menon et al., 2021; Hong et al., 2021). For the model $\Phi(\cdot)$, it is trained by the standard cross-entropy loss function on imbalanced training data set, and evaluated on test data. In this algorithm, the test logit is adjusted as follows:

$$\Phi(\boldsymbol{x}_n^t) = \Phi(\boldsymbol{x}_n^t) - \log p(y^s) \tag{6}$$

This simple procedure is derived from the Bayes optimal rule. It is apparent that Logit Adjustment involves a post hoc correction on an individual sample, which does not necessarily guarantee that the marginal distribution of the whole dataset matches the desired distribution.

## 3 METHODOLOGY

The first part of this section explores post-hoc correction from an OT perspective, proceeds to the automatic learning of the cost matrix via linear mapping. Lastly, we demonstrate how it can be achieved simply with one-layer neural network.

### 3.1 POST-HOC CORRECTION FORMALISED FROM AN OT PERSPECTIVE

Since Logit Adjustment applies adjustment at the individual sample level. It doesn't assure that the marginal distribution of the overall data set fulfils our desired distribution. In this respect, we clearly put the constraint into an equation:

$$\boldsymbol{Y}^{\intercal}\boldsymbol{1}_N = \boldsymbol{\mu}, \tag{7}$$

where $\boldsymbol{Y} \in \mathbb{R}^{N \times K}$ indicates the refined prediction value in matrix form, $\boldsymbol{\mu}$ represents the expected distribution on the test set. Alternatively, it is desirable to preserve another characteristic of $\boldsymbol{Y}$, namely, remaining almost as similar to the original prediction as possible. We consider inner-product based similarity to measure this, which is a straightforward yet useful similarity measure.

$$\underset{\boldsymbol{Y}}{\text{maximize}} \quad \langle C(\hat{\boldsymbol{Z}}), \boldsymbol{Y} \rangle, \tag{8}$$

where $\hat{\boldsymbol{Z}}$ represents the original prediction in matrix form, $C(\cdot)$ denotes to some transformation to $\hat{\boldsymbol{Z}}$, it can be some simple function, like Logarithmic function $\log(z)$, exponential function $z^\alpha$. Here we select $-\log(\cdot)$ as the cost function. This choice was driven by the requirement that the cost matrix must be positive definite, whereas the transformation of the original prediction by $-\log(\cdot)$ satisfies this condition. In addition, as log likelihood represents the local probability density of the associated samples, it can also be used to substitute $\hat{\boldsymbol{Z}}$ for the similarity approximation. In brief, the resulting numerical form can be put in formal terms as follows:

$$\underset{\boldsymbol{Y}}{\text{minimize}} \quad \langle -\log(\hat{\boldsymbol{Z}}), \boldsymbol{Y} \rangle \tag{9}$$

$$\text{subject to} \quad \boldsymbol{Y}^{\mathsf{T}} \mathbf{1}_N = \boldsymbol{\mu}, \quad \boldsymbol{Y} \mathbf{1}_K = \mathbf{1}_N. \tag{10}$$

Extra constraint on $\boldsymbol{Y}$ is imposed simply cos the tuned estimation has to fulfil the basic probabilistic requirement that its sum is one. Comparing Eq. (9-10) with Eq. (4), we can see that if we substitute $\boldsymbol{P}$ with $\boldsymbol{Y}$, and substitute $\boldsymbol{r}$ and $\boldsymbol{c}$ with $\mathbf{1}_N$ and $\boldsymbol{\mu}$ respectively, we find that the above optimisation problem is actually a special case of OT.

In preliminaries, the entropy regularised OT (EOT) is introduced. By adding entropy regularisation to OT, the given equation can be solved efficiently by Sinkhorn algorithm. Specifically, the equation is

$$\underset{\boldsymbol{Y}}{\text{minimize}} \quad \langle -\log(\hat{\boldsymbol{Z}}), \boldsymbol{Y} \rangle + \lambda \boldsymbol{Y} \log(\boldsymbol{Y}) \tag{11}$$

$$\text{subject to} \quad \boldsymbol{Y}^{\mathsf{T}} \mathbf{1}_N = \boldsymbol{\mu}, \quad \boldsymbol{Y} \mathbf{1}_K = \mathbf{1}_N.$$

The associated algorithmic flow for solving Eq. (11) is outlined in detail in Algorithm 1.

---

**Algorithm 1:** Solve OT-related algorithm efficiently in the post-hoc correction via Sinkhorn Algorithm.

---

**Input:** Cost matrix $\boldsymbol{M} = -\log(\hat{\boldsymbol{Z}})$, trade-off parameter $\lambda$, max number of iterations $N_T$, iteration number $t$, error threshold $\epsilon$, current error $\sigma$, row and column sums $\boldsymbol{r} = \mathbf{1}_N$ and $\boldsymbol{c} = \boldsymbol{\mu}$, $|\cdot|$ denotes the vector norm.

**Result:** Refined predictions $\boldsymbol{Y}$

1  Initialise $\boldsymbol{K} = e^{-\boldsymbol{M}/\lambda}, \boldsymbol{u}_{old} = \mathbf{1}_N, \boldsymbol{v} = \mathbf{1}_K, t = 0$;
2  **while** $t \leq N_T$ *and* $\delta \leq \epsilon$ **do**
3  $\quad \boldsymbol{u} = \boldsymbol{r}./\boldsymbol{K}\boldsymbol{v}$;
4  $\quad \boldsymbol{v} = \boldsymbol{c}./\boldsymbol{K}^{\mathsf{T}}\boldsymbol{u}$;
5  $\quad \sigma = |\boldsymbol{u}_{old} - \boldsymbol{u}|$;
6  $\quad \boldsymbol{u}_{old} = \boldsymbol{u}$;
7  $\quad t = t + 1$;
8  **end**
9  Output $\boldsymbol{Y} = \mathbf{diag}(\boldsymbol{u})\boldsymbol{K}\mathbf{diag}(\boldsymbol{v})$

---

Assigning $\lambda$ with 1, it is observed that we equate our objective function to the KL divergence, thus illustrating the extensive nature of our approach. DARP (Kim et al., 2020a) has previously applied it to long-tailed semi-supervised classification.

**Remark** We would like to illustrate the non-applicable scenarios of our method. Firstly, our method requires a large number of samples for evaluation. This is because if the batch size is small, we can not guarantee that the desired marginal distribution can be satisfied within the batch. In some online scenarios, the sample-wise correction method is more suitable. In addition, our method assumes that the marginal distribution is already known. We assume that it is consistent with a uniform distribution.

## 3.2 COST FUNCTION LEARNING VIA LINEAR MAPPING

Simple functions are likely to be sub-optimal for the real data sets; this suggests the design of a better cost function to better fit the long-tailed recognition problem. However, *manually designed cost functions require expert knowledge in different domains. Thus, we propose to use a linear mapping to automatically learn the cost function, which relieves the need to configuration.*

More specifically, for predictions $\tilde{Z}$ generated by Softmax operation via leveraging linear transformation matrix $W$, $W \in \mathbb{R}^{K \times K}$ is learned so that the following objective function is minimised.

$$\underset{Y \in \mathbb{R}}{\text{minimize}} \quad -\langle Y, \log \tilde{Z} \rangle + \lambda \langle Y, \log Y \rangle, \tag{12}$$

$$Y\mathbf{1}_K = \mathbf{1}_N \quad Y^\intercal \mathbf{1}_N = \mu, \quad \tilde{Z}_{nk} = \frac{\exp(W^\intercal \Phi(x_n))_k}{\sum_k^K \exp(W^\intercal \Phi(x_n))_k}.$$

The resulting formula can be illustrated using a simple one-layer network of weight parameter $W$, together with an error function in Eq. (12). We initialise $W$ with an identity matrix and use small learning rate to learn $W$. One could also absorb the term $-\log p(y^s)$ as the fixed bias parameter into the network. Motivated by this description, we can use a general gradient descent algorithm, such as SGD, to optimise the error function. Taking into account the implementation in practice, the direct calculation of Eq. (12) can be done using the Sinkhorn iterations (Burges et al., 2013; Frogner et al., 2015; Peyré et al., 2019) in mini-batch training efficiently. Besides, We term the proposed method as OTLM (*Optimal Transport via Linear Mapping*), Figure 1 illustrates the overview of OTLM.

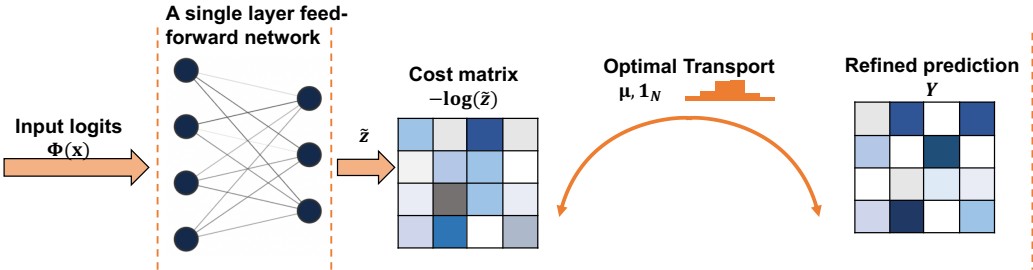

Figure 1: Our proposed framework OTLM. The logit $\Phi(x)$ is fed into a single-layer feed-forward network to infer $\tilde{Z}$. The cost matrix $M$ is set to $-\log \tilde{Z}$. By solving the optimal transport problem via Sinkhorn algorithm, we can obtain the refined prediction $Y$.

**Compute the gradients *w.r.t* $W$**   With the optimisation in Algorithm 1, it was conducted on the overall data set with a comparatively large number of data. Quite a different scenario now, as mini-batch training is more favoured when it comes to neural networks. For this reason, the optimisation workflow in Algorithm 1, as pointed by Peyré et al. (2019); Viehmann (2019), will have a problem of batch stablisation. To this end, we perform logarithmic transformation to step 3-4 in Algorithm 1.

$$\log u = \log r - \log(Mv) = \log r - \text{logsumexp}\left(\frac{1}{\lambda}M - \log v\right) \tag{13}$$

$$\log v = \log c - \log(M^\intercal u) = \log c - \text{logsumexp}\left(\frac{1}{\lambda}M - \log u\right). \tag{14}$$

with log-sum-exp operation $\text{logsumexp}(x) = \log(\sum_i \exp x_i)$. Assuming convergence is achieved in the Sinkhorn's loops, it is unnecessary for us to manually to compute the derivative of the loss function in Eq. (12) *w.r.t* $W$. As long as we make sure that the entire optimisation process is differentiable, modern deep learning libraries can automatically implement end-to-end derivations, as it allows us to differentiate between the results of Sinkhorn's loops as a mere composition of elementary operations.

Performance degradation caused by imbalanced distributions can be addressed with meta-learning based methods (Ren et al., 2020; Wang et al., 2020; Li et al., 2021). Recent works have illustrated neural networks can learn more meaningful representations from the validation data set $\{(x_n^v, y_n^v)\}_{n=1}^{N_v}$. We also take a validation data set to optimise the parameter $W$, *but with the still significant difference in that we require no labelling information, thus avoiding a large expanse of labeling samples.*

## 4 EXPERIMENTS

In this section, we first conduct experiments comparing our approach versus extant post-hoc correction methods on three data sets, including CIFAR-100-LT (Cao et al., 2019), ImageNet-LT (Liu et al., 2019), and iNaturalist (Horn et al., 2018) with varying backbones. Finally, we empirically make a comparison of our algorithm with alternative cutting-edge long-tailed recognition methods. Observe that, for one thing, *a plausible coupling of our algorithm to any training-aware long-tailed recognition method exists*. As an illustration of the potency and strong generalisation of our approach, we conducted post-hoc correction of the pseudo-predictions of both methods, which are fairly typical for the training stage of all data sets: The first is based on the *cross entropy (CE)*, and the other is the recently proposed *RIDE* (Wang et al., 2021), which is based on multi experts. *We conducted experiments in the appendix for semi-supervised learning, to some extent to imitate the online situation.* All our experiments are implemented in the PaddlePaddle deep learning platform.

### 4.1 DATA SETS AND BASELINES

The baseline methods and data sets are briefly described here, with implementation details placed in appendix. *We assume that the marginal distribution is uniform due to the characteristics of data sets.*

**Baselines**: We compare our methods with (i) post-hoc correction methods including Logit Adjustment (Menon et al., 2021), DARP (Kim et al., 2020a), (ii) state-of-the-art methods including Focal Loss (Lin et al., 2017), LDAM (Cao et al., 2019), BBN (Zhou et al., 2020), Balanced Softmax (Ren et al., 2020), Causal Norm (Tang et al., 2020), LADE (Hong et al., 2021), M2m (Kim et al., 2020b), Decouple (Kang et al., 2020), LFME (Xiang et al., 2020), RIDE (Wang et al., 2021)

**Long-tailed data set:** We take experiments on three data sets including CIFAR-100-LT, ImageNet-LT, and iNaturalist. We build the imbalanced version of CIFAR-100 by downsampling samples per class following the profile in Liu et al. (2019); Kang et al. (2020) with imbalanced ratios 10, 50, and 100. For all the benchmarks, we evaluate the performance on the test data set using a model trained on the training data set, and report the results using top-1 accuracy.

### 4.2 MAIN RESULTS

Table 1 illustrates the results on CIFAR-100-LT data set. For CE-based methods, it shows that Logit Adjustment is indeed a simple yet effective method for post-hoc correction. Remarkably, it outperforms the baseline methods by 2.6%, 4.1%, and 4.1% under the imbalance ratio of 10, 50, and 100 respectively. With a quick look at the results based on the OT approach, the superior results stress the advantages of our approach over the DARP and Logit adjustments. OT algorithm can further improve the performance by 0.62%, 0.43%, 0.70%, for OTLM, it outperforms Logit Adjustment by 0.55%, 0.73%, 0.99% under the three imbalance ratios on CIFAR-100-LT data set. For RIDE-based training-aware methods, there is also a 0.6% and 1.23% improvement in the accuracy of our method.

Table 1: Comparison on the top-1 accuracy with post-hoc correction methods on CIFAR-100-LT data set using ResNet-32 backbone, ImageNet-LT and iNaturalist data set using ResNeXt-50-32x4d and ResNet-50 backbones respectively. Better results are marked in bold, the small red font indicates the increase in accuracy when our method is compared to Logit Adjustment and RIDE. '-' means the results are not reported. CE indicates that a cross-entropy loss function is used in the training stage.

| Data set | CIFAR-100 | | | ImageNet-LT | iNaturalist |
|---|---|---|---|---|---|
| Imabalced Ratio | 10 | 50 | 100 | - | - |
| Baseline (CE) | 59.00 | 45.50 | 41.00 | 43.10 | 65.00 |
| Logit Adjustment+CE | 61.63 | 49.62 | 45.11 | 51.62 | 69.44 |
| DARP+CE | 61.78 | 49.52 | 45.13 | 51.20 | 71.18 |
| OT+CE | 62.25+0.62 | 50.05+0.43 | 45.81+0.70 | 51.87+0.25 | 71.20+1.76 |
| OTLM+CE | 62.18+0.55 | **50.35**+0.73 | **46.10**+0.99 | **52.35**+0.73 | **72.30**+2.86 |
| RIDE | - | - | 50.20 | 57.50 | 72.90 |
| OT+RIDE | - | - | 50.82+0.62 | 58.45+0.95 | 75.85+2.95 |
| OTLM+RIDE | - | - | **51.43**+1.23 | **58.71**+1.21 | **76.12**+3.22 |

Table 1 also provides the results on ImageNet-LT and iNaturalist data sets. The fact again attracts our attention that for CE-based training-aware methods, OT, which is based on convex optimisation, consistently outperforms Logit Adjustment by a large margin (1.76%) on iNaturalist data set. The accuracy boosting indicates the great potential of methods based on optimisation for post-hoc correction. Besides, not surprisingly, OTLM can further enhance the prediction accuracy, it outperforms Logit Adjustment by 0.73% and 2.86% on ImageNet-LT and iNaturalist respectively. As for RIDE, the gain in accuracy is also significant, about 3% on iNaturalist.

## 4.3 COMPARISON WITH THE STATE-OF-THE-ART METHODS

Armed with a demonstration of the validity of OT and OTLM, we switched to a comparison of our performance with existing methods that achieved the most advanced results on three benchmarks. As shown in Table 2, in particular, on ImageNet-LT and iNaturalist data sets, the experimental results are remarkable and impressive, our method outperforms RIDE by 1.0% and 3.0%, respectively. *Since iNaturalist is a very difficult and fine-grained data set consisting of 8,142 categories. These huge performance gains come at a fraction of the cost*, as we show in the following subsection.

Table 2: Comparison on the top-1 accuracy with state-of-the-arts on CIFAR-100-LT data set using ResNet-32 backbone, on ImageNet-LT and iNaturalist data set using ResNeXt-50-32x4d and ResNet-50 backbones respectively. − denotes the results are not reported, results underlined are the ones being compared, best results are marked in bold, the small red font denotes performance gain.

| Data Set | CIFAR-100 | | | ImageNet-LT | iNaturalist |
|---|---|---|---|---|---|
| Imabalced Ratio | 10 | 50 | 100 | − | − |
| Cross Entropy (CE) | 55.7 | 45.5 | 38.3 | 44.4 | 61.7 |
| Focal Loss | 55.8 | 44.3 | 38.4 | 43.7 | − |
| CB-Focal | − | − | − | − | 61.1 |
| OLTR | − | − | − | 46.3 | 63.9 |
| NCM | − | − | − | 47.3 | 63.1 |
| norm | − | − | − | 49.4 | 69.3 |
| cRT | − | − | − | 49.6 | 67.6 |
| LWS | − | − | − | 49.9 | 69.5 |
| LDAM | − | − | − | − | 64.6 |
| LDAM+DRW | 58.7 | 46.6 | 42.0 | − | 68.0 |
| BBN | 59.1 | 47.0 | 42.6 | − | 69.3 |
| Causal Norm | 59.6 | 50.3 | 44.1 | 51.8 | − |
| M2m | 58.2 | − | 43.5 | − | − |
| LFME | − | − | 43.8 | − | − |
| Balanced Softmax | 61.6 | 49.9 | 45.1 | − | − |
| LADE | 61.7 | 50.5 | 45.4 | 51.9 | 70.0 |
| RIDE (6 experts) | − | − | 50.2 | 57.5 | 72.9 |
| Logit Adjustment | 61.6 | 49.6 | 45.1 | 51.6 | 69.4 |
| DARP | 61.8 | 49.5 | 45.1 | 51.2 | 71.2 |
| OT+CE | **62.3**$_{+0.7}$ | 50.1 | 45.8 | 51.9 | 71.2 |
| OTLM+CE | 62.2 | **50.4**$_{+0.8}$ | 46.1 | 52.4 | 72.3 |
| OT+RIDE (6 experts) | − | − | **50.8**$_{+0.6}$ | **58.5**$_{+1.0}$ | **75.9**$_{+3.0}$ |
| OTLM+RIDE (6 experts) | − | − | **51.4**$_{+1.2}$ | **58.7**$_{+1.2}$ | **76.1**$_{+3.2}$ |

## 4.4 COMPUTATION COST OF OT AND OTLM

As we have highlighted, the additional computational cost of OT and OTLM is particularly small, compared to that of training-aware methods. In Table 3, exact time of the evaluations on the ImageNet-LT and iNaturalist data sets is provided. Please note, the time here are measured from the start of each method to the final best performance. For the comparison between OT and OTLM, because OTLM runs on the GPU, one optimisation iteration of OTLM is in fact much faster than OT. Except for OTLM, which was run on an NVidia card (V100), the results come from a 28-core machine (2.20 Ghz Xeon). *Firstly, we can observe that all post-hoc correction methods here are not quite time-consuming*. For comparison, ImageNet-LT and iNaturalist can be trained on 4 NVidia cards

Table 3: Time for different methods to execute on the ImageNet-LT and iNaturalist data sets. The times here are measured from the start of each method to the final best performance. Their running time are counted in seconds. Coupled with the prior results, an observation can be made that OT can be the best coordinate to trade off performance and efficiency in post-hoc correction methods.

| Data Set | ImageNet-LT | iNaturalist |
|---|---|---|
| Logit Adjustment | 0.3 | 1.2 |
| OT | 32.7 | 82.5 |
| OTLM | 134.4 | 256.5 |

(V100) for extremely long periods of time. Using ResNeXt-50-32x4d on the iNaturalist training data set, for example, with a batch size of 256, the amount of training time (in seconds) to perform 1 iteration is approximately 850. *Also, if one differs the performance of each method, we gather that OT and OTLM provide the best balance of performance and running costs to match.*

### 4.5 CONVERGENCE BEHAVIOR OF OTLM

To verify that our adopted OTLM algorithm is really efficient. With ResNeXt-50-32x4d as the backbone, we plotted curves depicting the decreasing absolute error $\sigma$ with the number of iterations $t$ on the ImageNet-LT validation data set. Image on the left in Figure 2 gives us a visualisation of the optimisation process. It is found that the absolute error $\sigma$ converges at around 60 iterations. In fact, we discovered that 10 or even fewer iterations were enough to yield satisfactory performance.

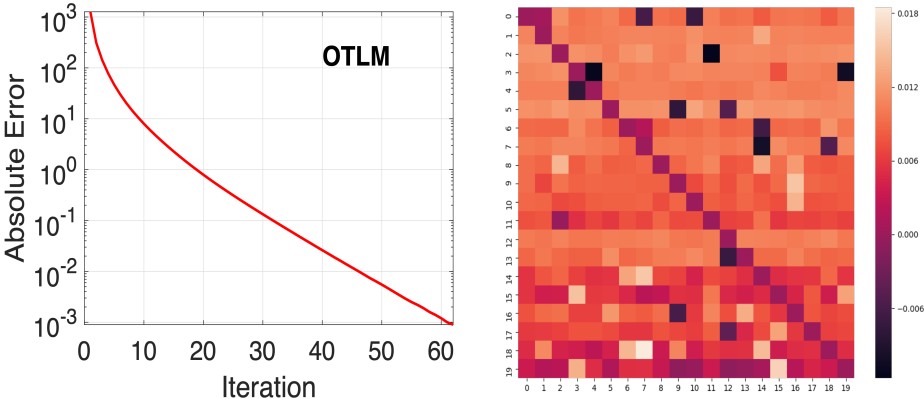

Figure 2: The left image shows the convergence of OTLM: The absolute error $\sigma$ decreases with iterations on the ImageNet-LT validation data set (20k samples), and we used ResNet-50 as the backbone to obtain the original predictions $\boldsymbol{Z}$. The right image demonstrates the block in $\boldsymbol{W}$, we trained on CIFAR-100 to produce the original predictions with OTLM to minimise the difference between the ideal and actual distributions. For ease of display, the values of the diagonal elements are relatively large and we will replace these values with zeros.

### 4.6 DETAILED ANALYSIS ON DISTRIBUTION MATCHING

Upon distribution fitting, preferably the estimation bias is lessened, which implies that the KL divergence distance between the refined prediction and the target distribution is decreased. Table 4 displays the KL distances on the ImageNet-LT and iNaturalist data sets, and we can find that OT possesses a smaller KL distance than Logit Adjustment, which suggests that OT decreases the estimation bias more than Logit Adjustment by a much larger margin.

### 4.7 VISUALISATION ON WEIGHT PARAMETER IN OTLM

As a natural consequence, one wonders about the exact structure of this weight matrix $\boldsymbol{W}$, and no better way to present its results than through visualisation. Towards this end, we applied OTLM

Table 4: KL distance between refined probability distribution and target distribution computed on ImageNet-LT and iNaturalist data sets. Smaller distance indicates two distributions are more similar.

| Data Set | ImageNet-LT | iNaturalist |
|---|---|---|
| Logit Adjustment | 1.78e-05 | 5.81e-06 |
| DARP | 8.49e-08 | 1.07e-08 |
| OT | 8.62e-10 | 4.72e-10 |

to refine the original predictions on CIFAR-100 data set with an imbalanced ratio $R = 100$ and ResNet-32 backbone, and trained a one-layer model until it converged to extract the value of $\boldsymbol{W}$, where the shape of is $100 \times 100$. We visualised a sub-block of size $20 \times 20$, the illustration is presented on the right image in Figure 2. The visualised matrix is very dense, this non-sparsity suggests that the OTLM manages to correct the predictions of the model in the training stage, whilst minimising the distance between the desired and the actual distribution.

## 5 RELATED WORK

**Long-tailed recognition:** The great majority of current methods applied to long-tailed recognition can be divided into either training-aware and post-hoc correction methods. The training-aware methods aim to improve the generalisation ability of models trained on imbalanced training data set. Among them, re-sampling and re-weighting are two major methods. Re-sampling controls the class numbers by the means of over-sampling (Chawla et al., 2002) and under-sampling (Barandela et al., 2004; Drummond et al., 2003). The re-weighting approaches achieve balanced learning by giving smaller weights to majority classes and larger weights to minority classes (Tang et al., 2008; Zadrozny et al., 2003; Lin et al., 2017; Lee et al., 2017; Cui et al., 2019; Khan et al., 2019). Other approaches have also been proposed to resolve the imbalanced problem. For example, Cao *et al.* introduces a theoretically-principled loss function motivated by minimizing the generalisation error bound (Cao et al., 2019), Kang *et al.* proposes to decouple the learning phases into representation learning and classifier tuning (Kang et al., 2020). Apart from training-aware methods, post-hoc correction also offers us an avenue for long-tailed recognition problem, it can be applied to the normalisation of classifier weights (Kang et al., 2020; Zhang et al., 2019; Kim & Kim, 2020) or logits based on label frequency (Provost, 2000; Collell et al., 2016). Among them, Logit Adjustment (Menon et al., 2021) is a simple yet powerful method to use. Compared with Logit Adjustment, our approach performs the post-hoc correction from the optimisation view.

**Optimal transport:** OT distance is a important family of distances for probability measures (Villani, 2008; Santambrogio, 2015). It has been extensively applied in different fields in machine learning. Courty *et al.* (Courty et al., 2016) proposes a regularised unsupervised optimal transportation model to perform the domain adaption. Other applications include 3D shape matching (Su et al., 2015), generative model (Arjovsky et al., 2017; Salimans et al., 2018; Bunne et al., 2019; Deshpande et al., 2019; Bhagoji et al., 2019; An et al., 2021), graph matching (Xu et al., 2019a;b), model designs Kandasamy et al. (2018); Chizat & Bach (2018). To the best of our knowledge, we are the first to perform the post-hoc correction from the OT perspective.

## 6 CONCLUSION

In this paper, we present two techniques for post-hoc corrections from an optimisation perspective. Our approach reviews the basic mathematical formulation of the distribution alignment problem and relates the resulting formulation to OT, which we find, moreover, offers greater possibilities and flexibility to solve the problem. In order to avoid manual configuration of the cost matrix, we propose to learn the cost matrix automatically by linear mapping, which leads to further performance improvements. The experimental results confirm the superiority of our proposed approach. We believe our work can open up more possibilities and imagination for post-hoc corrections. There are potential challenges that remain to be addressed. These include, for example, how to apply OT to correction of predictions under unknown marginal distribution. Meanwhile, an extension of our approach to other applications where label shift is involved is considered possible.

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

# A  OPTIMAL TRANSPORT ON LONG-TAILED SEMI-SUPERVISED CLASSIFICATION

Together, with the extension of our approach, we adapt it to semi-supervised long-tailed classification tasks. We bring our approach together with the advanced semi-supervised learning methods FixMatch Sohn et al. (2020). Adhering to the configuration in DARP Kim et al. (2020a), we also create synthetically long-tailed variants of CIFAR-100 with imbalanced ratio $R = 20$. The sample size of the unlabeled data is twice as large as the labelled data, with the same proportion of imbalance between them. To assess the performance, we report two popular metrics: balanced accuracy (bACC) Huang et al. (2016) and geometric mean scores (GM) Branco et al. (2016). Experimentally, with the exception of an OT to align the marginal distribution of the unlabelled data, all settings are consistent with DARP, and we set $\lambda$ to 0.005. The results are reported in Table 5. It can be seen that our approach also achieves much better performance on semi-supervised learning tasks. *It can somehow mimic the online scenarios. And the advantages of OT coping with such cases can be demonstrated.*

Table 5: Comparison of classification performance (bACC/GM) on CIFAR-100 with imbalanced ratio $R = 20$, We report standard deviation and mean values for each evaluation metric. Better results are marked in bold.

| Method | CIFAR-100 | |
|---|---|---|
| | bACC | GM |
| DARP | $54.9_{\pm 0.05}$ | $46.4_{\pm 0.41}$ |
| OT | $\mathbf{56.6}_{\pm 0.08}$ | $\mathbf{48.4}_{\pm 0.32}$ |

# B  PARAMETER STUDY OF $\lambda$

We conduct an additional experiment on iNaturalist data set to study the effect of $\lambda$ on the final performance. The value of $\lambda$ is selected from $\{0.01, 05, 0.1, 1.0.2.0, 5.0\}$, all other experimental settings are the same, including various hyperparameters, network structure, etc. As illustrated in Figure 3, we can find that OT achieves the best performance when $\lambda = 1$.

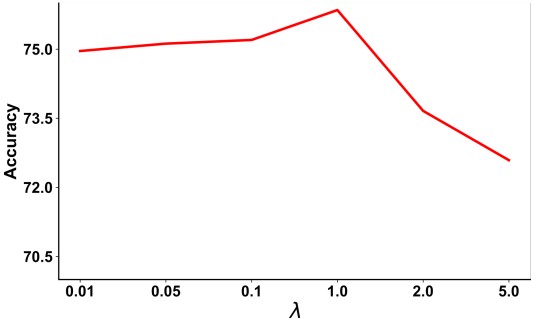

Figure 3: Visualisation of the accuracy change curve with $\lambda$ on iNaturalist data set.

# C  IMPLEMENTATION DETAILS

The specific implementation details for each data set under the different methods are described below.

## C.1  IMPLEMENTATION DETAILS OF CROSS ENTROPY

**CIFAR-100-LT**: During the training stage, following the recipe of (Tang et al., 2020; Hong et al., 2021), we apply SGD with batch size 256 and weight decay 0.0005 to train a ResNet-32 (He et al., 2016) model for 200 epochs, we employ the linear warm-up learning rate schedule for the first five

epochs. We also set the base learning rate to 0.2 and reduce it at epoch 120 and 160 by a factor of 100. During the post-hoc stage, we use Adam with batch size 10k and weight decay 0.001 to train the one-layer feed-forward network for 20 iterations. We set $\lambda$ to 0.1, $T$ to 200, $\Delta$ to 0.001. The learning rate is constant and set to 0.001.

**ImageNet-LT**: During the training stage, following the implementation of (Hong et al., 2021), we train the ResNeXt-50-32x4d (Xie et al., 2021) for 90 epochs and perform a cosine learning rate scheme with an initial learning rate of 0.05. SGD is also employed to optimise the neural network with batch size 256, weight decay 0.0005, and momentum 0.9. During the post-hoc stage, we use SGD with batch size 10k, weight decay 0.001 to train the one-layer feed-forward network for 100 iterations. The constant learning rate is 0.2. We set $\lambda$ to 1.2, $T$ to 7, $\Delta$ to 0.001.

**iNaturalist**: During the training stage, ResNet-50 (He et al., 2016) is chosen as the backbone network. We use SGD with momentum 0.9, batch size 256 to train the network. We utilise the cosine learning rate schedule gradually decaying from 0.1 to 0. During the post-hoc stage, we use SGD with batch size 23,826, weight decay 0.001, $\lambda$ to 1.0, $\Delta$ to 0.001 and $T$ to 10. The constant learning rate is set to 0.05. As iNaturalist has no test data set, we directly report the performance on validation data set.

## C.2 IMPLEMENTATION DETAILS OF RIDE

The training details we use in RIDE share the same as the original paper, there are 6 experts in our RIDE on all three data sets.

**CIFAR-100-LT** We experimented with $R = 100$ and ResNet-32 as the base network. SGD is utilised as our solver with momentum 0.9, batch size 128, and epoch 200. The learning rate is initialised as 0.1 and decayed by 0.01 at epoch 120 and 160 respectively.

**ImageNet-LT** We take experiments with ResNeXt-50 as backbone with 100 epochs. We set the initial learning rate to 0.1 and reduce it by 10 at epoch 60 and 80. SGD with batch size 256 and momentum 0.9 is also adopted as the optimisation solver here.

**iNaturalist** We conduct experiments with ResNet-50 as backbone. The training details are the same as ImageNet-LT except from batch size 512.

