# OpenReview forum: "Optimal Transport for Long-Tailed Recognition with Learnable Cost Matrix"
_ICLR.cc/2022/Conference — ICLR 2022 Poster_

### Official Review · Reviewer_uP2E · 2021-11-01

**Correctness:** 3
**Technical Novelty And Significance:** 3
**Empirical Novelty And Significance:** 3
**Recommendation:** 6
**Confidence:** 3

**Main Review:**

Strength:

1. Post-hoc adjusting the prediction via OT is novel.
2. The proposed method is technically sound. By accurately aligning the predicted label distribution with the balanced label distribution (Table 4), the paper successfully outperforms the state-of-the-art post-hoc adjustment methods in terms of recognition accuracy (Table 1, 2).
3. A convergence analysis for the proposed OTLM is provided (Sect 4.5), making the paper easier to be reimplemented and reproduced.

Weakness:

1. The described post-hoc correction may not be feasible in real-world applications.

   In Sect 3.1, the authors assume that the "expected distribution on the test set" is known, followed by an explicit constraint $Y^T1_N=c$ for the optimization. In another word, the paper assumes the test set to have a balanced label distribution and optimize the predictions towards the balanced label distribution.

   However, the assumption is not always true. The test set in many long-tailed datasets is not balanced, e.g., the LVIS dataset. Those datasets use balanced metrics, e.g., mAP, to measure the model performance instead. In this case, assuming that the test set's label distribution is known might be improper.

   A follow-up question would be how to employ the method in real-world applications, where the test set could be small? If we try to make a single-image prediction, it may not seem possible to conduct the proposed OT optimization on only one data point. How should the method adapt to these scenarios?

   Note also that the proposed method takes significantly longer inference time than compared methods (Table 3).

2. Some empirical analysis is not sufficient. 1) In both Table 1 and Table 2, OTLM+RIDE seem to be missing. Can OTLM combine with ensemble-based methods? 2) What is the effect of the entropy regularization coefficient $\lambda$ in Equation 9? There lacks of an ablation analysis.

3. Some minor errors: 1) "...how it can be achieved simply one-layer neural network." -> "...how it can be achieved simply by a one-layer neural network." 2) "As a matter of fact, as long as we make sure the entire optimization process is differentiable." The sentence seems to be unfinished. 3) "DAPR" in both Table 1 and Table 2 should be "DARP". 4) Table 1 and Table 2 substantially overlap with each other. They have inconsistent precisions as well. Merging them into one table could make it clearer. 5) Figure 1 is a little misleading. Should the feedforward network have the same amount of input and output (four in the case)?





**Summary Of The Paper:**

This paper proposes to solve the long-tailed recognition by aligning the predicted label distribution with the test label distribution via Optimal Transport (OT).

**Summary Of The Review:**

I recommend a borderline reject. Although using OT is a novel idea, my major concern is that it could be infeasible to employ the OT-based method in practice. Thus, I am unsure if the new direction is worth further exploration.

=========== after reading author's latest revisions======

I find the limitations of the methods adequately addressed in the latest revision. I agree with the authors that this paper's setting is consistent with some existing literature and the proposed method can be useful in some offline applications after reading the author's feedback. I would like to raise my recommendation from borderline reject to borderline accept.

---

> ### Author Response · Authors · 2021-11-12
> **Response to reviewer uP2E**
>
> Many thanks to reviewer for your constructive feedback. We are encouraged that you found our method to be technically sound. Meanwhile, we are glad that you discovered that our experimental results successfully outperform the state-of-the-art post-hoc adjustment methods in terms of recognition accuracy. We address your comments below and will incorporate all feedback in the revision.
>
> ### Q1:  The test set in many long-tailed datasets is not balanced, e.g., the LVIS dataset.
>
> The balanced test set is only a special case, and in fact our approach is just as applicable as long as we know the marginal distribution of the test set. And this setting is recognized in many papers [1,2,3] in this field, and is also performed in their experiments.
>
> ### Q2: Assuming that the test set's label distribution is known might be improper.
>
> **Consider that the uniformity of the distribution on the validation and test data sets, it could be treated as an implicit prior knowledge to be applied**, exactly as the previous approach did, such as Balanced Softmax [2], Logit Adjustment [1]. Stepping back, even if we do not know the marginal distribution of the test dataset in advance. There are still ways to estimate the marginal distribution of the test dataset relatively precisely, such as methods in [4] and [5].
>
>
>
>
>
> ### Q3: how to employ the method in real-world applications, where the test set could be small?
>
> When evaluating, we do need to assume a relatively large data set. This is because if the sample size is relatively small, we do not always have the desired marginal distribution. In this case the sample-wise correction can be applied.
>
> However, we believe that **this is a very rare situation in reality**, which can be seen as an extreme case. That is, there is a large amount of data for the training sample, but a relatively small number of test samples. This is because in reality, machine learning models need to be validated on a large test data set.
>
>
>
>
>
> ### Q4: The proposed method takes significantly longer inference time than compared methods
>
> Our method does require a longer inference time, but **we do not consider this time increase to be significant and we consider this time to be acceptable**. Since OT and OTLM can provide better performance than logit adjustment, we gather that OT and OTLM provide the best balance of performance and running costs to match the balance between effectiveness and efficiency.
>
>
>
> ### Q6: Can OTLM combine with ensemble-based methods?
>
> We have supplemented our experimental results with OTLM+RIDE, which show that the combination of our OTLM and RIDE can further improve performance.
>
>
>
> ### Q7: What is the effect of the entropy regularization coefficient.
>
> For any $\lambda>0$, the solution $\boldsymbol{P}^\lambda$ is unique and has a closed form solution.  Besides, because of the entropy regularization, $d_{\boldsymbol M}^{\lambda}$ is necessarily larger than $d_{\boldsymbol M}$. The entripy regularization can also be regarded as  considering maximum-entropy principle. We have included an additional experiment in the appendix to study the effect of $\lambda$ on iNaturelist data set.
>
>
>
> ### Q8: Some minor errors
>
> Thank you so much, we have fixed some grammatical errors, redrawn the images, and completed some sentences that seemed unfinished in the updated revision.
>
> [1] Long-tail learning via logit adjustment
>
> [2] Balanced Meta-Softmax for Long-Tailed Visual Recognition
>
> [3] BBN: Bilateral-branch network with cumulative learning for long-tailed visual recognition
>
> [4] Using trusted data to train deep networks on labels corrupted by severe noise.
>
> [5] Regularized learning for domain adaptation under label shifts

---

> > ### Comment · Reviewer_uP2E · 2021-11-12
> > **Concern about this paper's setting**
> >
> > Thank you for your clarifications. However, my concern in Q1 remains.
> >
> > > The balanced test set is only a special case, and in fact our approach is just as applicable as long as we know the marginal distribution of
> > > the test set. And this setting is recognized in many papers [1,2,3] in this field, and is also performed in their experiments.
> >
> > I can't agree with the argument. In fact, this paper may have misunderstood previous literature's setting. In Sect 2. [1], there is a discussion on why we should use $P^{bal}(y|x)$ for inference:
> >
> >     "To cope with this, a natural alternative is the balanced error, which averages each of the per-class error rates... This can be seen as implicitly using a balanced class-probability function...".
> >
> > This implies that the test set may not be balanced as long as a balanced error is used. For example, we can have different numbers of samples in each class in the test set, Logit Adjustment can still be applied as long as we use a mean error rate over all classes. Therefore, [1] does not actually assume that the test set has to be balanced. Instead, [1] assumes a **balanced error** is used. [2, 3] use the same assumption as well.
> >
> > Thus, there exists a difference between the setting in this paper and the setting in previous long-tailed learning literature: this paper explicitly requires a balanced test set for optimization. In my opinion, this assumption can be too strong in practice: how can we collect a perfectly balanced test set when data is naturally distributed and we don't know their labels?

---

> > > ### Author Response · Authors · 2021-11-12
> > > **Response to concern about this paper's setting**
> > >
> > > Thank you very much for your reply, and we are glad that some of our previous answers have helped you to resolve some of your concenrs and confusion. In response to the paper on whether the a priori knowledge of the marginal distribution of the test set is known. We would like to elaborate further.
> > >
> > > We agree with you that we can not collect a perfectly balanced test set when data is naturally distributed.
> > >
> > > First, in the paper [1],you can see that the authors use this a priori knowledge implicitly when they say the following:
> > >
> > > ***This can be seen as implicitly using a balanced class-probability function $P^{bal}(y|x)\propto \frac{1}{L}P(x|y)$***
> > >
> > > In the paper [2], in the derivation of Theorem 1. The authors make the following assumptions.
> > >
> > > ***Assume $\phi$ to be the desired conditional probability of the balanced dataset, with the form $\phi_j=p(y=j|x)=\frac{p(x|y=j)}{p(x)}\frac{1}{k}$***
> > >
> > > As can be seen, they both assume that class-probability is an implicit knowledge. This is exactly what we have assumed in our paper. *The uniformity of the test set is also only a very special assumption we make for the three datasets CIFAR-100, iNaturelist, and ImageNet-LT*. Specifically, our method can be applicable as long as we know the marginal distribution of the test set. So the core issue is i.e., **assuming we don't know the marginal distribution of the test set, how do we go about estimating it?**
> > >
> > > In fact, there is a lot of literatures [3,4] that have addressed this issue very well. We consider this concern of yours to be also a very open and urgent issue to be addressed in the future.
> > >
> > >
> > > [1] Long-tail learning via logit adjustment
> > >
> > > [2] Balanced Meta-Softmax for Long-Tailed Visual Recognition
> > >
> > > [3] Using trusted data to train deep networks on labels corrupted by severe noise.
> > >
> > > [4] Regularized learning for domain adaptation under label shifts

---

> > > > ### Comment · Reviewer_uP2E · 2021-11-13
> > > > **Clarification on the argument about the assumption**
> > > >
> > > > Thank you for your further explanation.
> > > >
> > > > I would clarify that my argument in the previous comment was: [1, 2, 3] make the assumption not because their test set is really balanced, but because they use a **balanced error** as the evaluation metric.
> > > >
> > > > [1, 2] can approximate $P^{bal}(y|x)$ even if there is not a balanced test set. For example, [2] provides experiments on LVIS, whose train set and test set are from the same long-tailed distribution, and a balanced evaluation metric (mAP) is used. On the other hand, this paper will not be applicable to LVIS. This is an important limitation of the proposed method, and the limitation results from the difference in assumptions.
> > > >
> > > > I agree with the authors that there are research works trying to adapt to arbitrary known test distributions, for example, [4]. However, since this paper mainly compares with [1], I would recommend the authors follow [1]'s setting or make the above-mentioned limitation clear in the manuscript.
> > > >
> > > > [1] Long-tail learning via logit adjustment
> > > >
> > > > [2] Balanced Meta-Softmax for Long-Tailed Visual Recognition
> > > >
> > > > [3] BBN: Bilateral-branch network with cumulative learning for long-tailed visual recognition
> > > >
> > > > [4] Disentangling Label Distribution for Long-tailed Visual Recognition

---

> > > > > ### Author Response · Authors · 2021-11-13
> > > > > **Add a remark about the assumption in this paper**
> > > > >
> > > > > Thank you very much for the further clarification.
> > > > >
> > > > > We have checked these previous articles and it's indeed from the perspective of a **balanced error**. We have added a remark about the assumption of our paper in section 3.1 in the latest revision.

---

### Official Review · Reviewer_vjwX · 2021-11-02

**Correctness:** 3
**Technical Novelty And Significance:** 3
**Empirical Novelty And Significance:** 2
**Recommendation:** 6
**Confidence:** 3

**Main Review:**

General comments:
The overall idea proposed by the paper is interesting and seems well-motivated. Though applying optimal transport to deal with imbalanced data is not particularly new[1], the proposed method in a post hoc manner is generally more flexible. With extensive experiments, the method is demonstrated to be effective.

Strengths:
1. The paper is self-contained and the main idea can be easily grasped. The introduction clearly states the research question and the motivation of the work.
2. Formalizing the post hoc correction from an OT perspective seems interesting and well-justified. How the authors draw the connection between OT and logit adjustment is particularly appealing to me.

However, some substantial concerns should be addressed before the paper can be accepted.

Weaknesses or Concerns:
1. Does the paper inherently assume that the batch size during evaluation is rather large? I am asking this because the constraint in equation (5) may fail if there are only a few images (e.g. 2 images) in a batch. In practice, it is unlikely that we can always have a batch with the same distribution as the desired one. Sample-wise correction may have its downsides, but it fits in most setups.
2. The training of OTLM seems to depend on the convergence of Sinkhorn’s algorithm. I am wondering how long would it take to train the OTML? For a post hoc approach, we generally expect it to be flexible. If the post hoc method involves learning from the data, I am not sure whether OTLM can generalize from one dataset to another.
3. It seems to me that the variant, OTLM, is not well-justified. From Table 2 and Table 3, I notice that the performance of OTLM is not much different from OT(sometimes even worse) but with a significantly slower inference speed. I would be thankful if the authors could elaborate more on why do we need linear mapping and what has it learned.

Minor Issues:
1. Why the evaluation of OTLM is much slower than OT even on GPU? Isn’t there only one layer of feed-forward network?
2. ‘our method outperforms by RIDE by 1.0% and 3.0%’ -> sounds a bit over-claimed to me since the results obtained by coupling RIDE.
3. For iNaturalist, the training and evaluation of OTLM are both done on the validation set. It seems like an unfair comparison to me.

[1] Yan et al., Oversampling for Imbalanced Data via Optimal Transport, AAAI 2019

**Summary Of The Paper:**

This paper contributes an extension of post hoc correction of long-tailed recognition with Optimal Transport (OT). Unlike the previous work (e.g. logit adjustment) which focuses on sample-wise correction, this work, on the other hand, considers the marginal distribution of the overall data for correction. The method is further extended by learning a cost matrix. Their experiments show the effectiveness of optimal transport in the long-tail recognition problem via comprehensive comparison with previous works in terms of performance and efficiency.

**Summary Of The Review:**

Despite some drawbacks on the assumption made and experiment evaluations, the paper would still benefit the community. I therefore lean towards a positive direction at this moment.

Update:  As mentioned by reviewer uP2E, the strong assumption of knowing the marginal distribution of labels is not desirable in real-world scenarios. The proposed approach can be only applied to offline applications. I would recommend the authors include the following items in the future version:

1. The performance of the proposed approach with an estimated marginal distribution;
2. How can one apply the proposed method to online scenarios? (e.g. store evaluated samples to form a large batch)

---

> ### Author Response · Authors · 2021-11-12
> **Response to reviewer vjwX**
>
> We thank the reviewer for your constructive comments. We are encouraged that you discovered that our proposed method in a post hoc manner is generally more flexible. In addition, we are glad that you found our approach to be evaluated with adequate experiments and achieved significant improvements when compared against baselines. We answer your questions below, and we will incorporate all feedback in the revision.
>
> ### Q1: Does the paper inherently assume that the batch size during evaluation is rather large?'
>
>  Yes, when evaluating, we do need to assume a relatively large batch size. This is because if the sample size is relatively small, as in the extreme case you have given (e.g. 2 images), we do not always have the desired marginal distribution in the batch. In this case the sample-wise correction can be applied. In short, our method provides better adjustment when the batch size is large.
>
> However, we believe that **this is a very rare situation in reality**, which can be seen as an extreme case. That is, there is a large amount of data for the training sample, but a relatively small number of test samples. This is because in reality, machine learning models need to be validated on a large test data set.
>
>
>
> ### Q2: how long would it take to train the OTML
>
> In Table 3, We give the running times of OTLM on ImageNet and iNaturelist as 134.4 and 256.5 seconds respectively.
>
>
>
> ### Q3: For a post hoc approach, we generally expect it to be flexible
>
> In fact, we consider OTLM to be a very flexible approach. For OT, choosing the exact cost matrix when applying OT is challenging and requires expert knowledge of various tasks. The data varies considerably between different tasks. Even for the same task, it is possible to have extremely different data sets, so it is reasonable to learn a better cost matrix from the data.
>
>
>
> ### Q4: why do we need linear mapping
>
> For OT, for better learning of the cost matrix, we present a linear mapping enabling very elegant learning with one-layer neural network, it can also avoid unnecessary configurations and expert knowledge.
>
>
>
> ### Q5: The performance of OTLM is not much different from OT(sometimes even worse) but with a significantly slower inference speed
>
> In fact, if we look at table 1, we see that the advantages of OTLM over OT can be even more pronounced on larger data sets. For example, on iNaturelist data set which contains more than 8,000 classes, OTLM can further enhance the prediction accuracy,  it outperforms OT by 1.1%. The boost on such complex data is impressive.
>
>
>
> ### Q6: Why the evaluation of OTLM is much slower than OT even on GPU？
>
> Note that in Table 3, the time for OTLM is the time from the start of training to the final better performance, not the time for one iteration of OTLM. Because OTLM requires a certain number of iterations to find a better cost matrix which consequently leads to a better performance. In fact, OTLM is much faster than OT in a single iteration because it runs on the GPU.
>
> ### Q7: Sounds a bit over-claimed to me since the results obtained by coupling RIDE.
>
> The point we are trying to make here is that our method can be mixed with various training-aware methods and achieve further performance gains. In fact, we are very grateful to the RIDE algorithm for the significant improvement in the performance of the model during the training stage, and our approach stands on the shoulders of "this giant" for further performance improvements.
>
>
>
> ###  Q8: the training and evaluation of OTLM are both done on the validation set
>
> Since iNaturelist does not have a test set, so we did this, but on the ImageNet-LT dataset we validated on validation and tested on the test set. And in this field, many meta learning based papers also follow this practice. For iNaturelist, they also need to use the validation data set, but we only need the input samples without corresponding labels.

---

> > ### Comment · Reviewer_vjwX · 2021-11-13
> > **Thanks for the clarification**
> >
> > Thanks for the clarification.
> >
> > **Q2: how long would it take to train the OTLM**
> > > Note that in Table 3, the time for OTLM is the time from the start of training to the final better performance, not the time for one iteration of OTLM.
> >
> > > In Table 3, We give the running times of OTLM on ImageNet and iNaturalist as 134.4 and 256.5 seconds respectively.
> >
> > It would be better if the paper states how the evaluation time is measured clearly. The caption of Table 3, 'Time for different methods to execute on the ImageNet-LT anxd iNaturelist datasets.' is quite ambiguous and misleading since OTLM requires training while OT does not.
> >
> > **Q1: Does the paper inherently assume that the batch size during evaluation is rather large?'**
> > >However, we believe that this is a very rare situation in reality, which can be seen as an extreme case.
> >
> > On the contrary, I think this is rather common. A great number of test samples is not equivalent to a large batch size during evaluation. One can find a counter-example easily. In most of the online applications, it seems the method proposed would not be a good fit. For example, 3D detection for autonomous driving.
> >
> > Typo:
> > iNaturelist  -> iNaturalist

---

> > > ### Author Response · Authors · 2021-11-13
> > > **Description of running time, correction of typo, description of the applicability of our method**
> > >
> > > Thank you very much for the further elaboration.
> > >
> > >  Yes, we realise that in real life there are many such situations. For example, in an online scenario, the number of samples is sometimes relatively small and sample-wise correction is more often required. We have added a remark to clarify this situation in section 3.1 in the updated revision.
> > >
> > > We have fix the typo in our revision.  iNaturelist -> iNaturalist

---

> ### Author Response · Authors · 2021-11-26
> **Assumption of a balanced test set**
>
> Thank you very much for your latest response, we would like to supply further clarification on this assumption.
>
> First, our approach is indeed more suitable for offline applications. We also consider that it is unlikely that a perfectly balanced dataset will be available in the real world. In practice, however, **our method can also be applied to an imbalanced data set, provided we know the marginal distribution of that data set**. **Specifically, in our latest revision, we can assign $\boldsymbol \mu$ to the ideal distribution that we desire in equation10, not limited to the balanced distribution**. And there are a number of methods that can be used to estimate marginal distributions [1-2].
>
> Further, the assumption of balanced test set we make here is due to the characteristics of the data sets in our experiments. **Because the CIFAR-100, ImageNet-LT and iNaturalist datasets all have balanced test sets, however, our approach is also applicable in the case of imbalanced test sets**. Also, as reviewer uP2E agreed, this paper's setting is also consistent with some existing literature.
>
> In addition, our approach can be highly applicable in offline scenarios, especially in the context of large data. The experimental results show the advantages of our approach over other post-hoc correction methods. Furthermore, our methods can be freely coupled with various training-aware methods.
>
> [1] Using trusted data to train deep networks on labels corrupted by severe noise.
>
> [2] Regularized learning for domain adaptation under label shifts.

---

> ### Author Response · Authors · 2021-11-27
> **Online application of our method, and performance of the method under the estimated marginal distribution.**
>
> So far, we really appreciate your interest in our paper. Thank you for the great efforts and time you have devoted to our work. You are really one of the best reviewers we have ever met. You raise excellent questions that keep us engaged in an ongoing process of thinking about and coming up with solutions to problems. In particular, your latest question has led to a scenario where the application of our paper may indeed become possible for online implementation. Many thanks for your comments and for the inspiration you have brought to our work.
>
> ### **Online application of our method**
>
> In the discussions that have preceded this, the reviewers and we have come to an agreed conclusion: our approach is more adaptable to the offline scenario. Because the sample size online may be relatively small, then the marginal distribution of small volumes of data does not fulfill the target distribution.
>
> However, The latest suggestion you provided, i.e., **perhaps we can store evaluated samples to form a large batch. This is, in our mind, a perfectly feasible solution**. In fact, **this scheme has been used in many other areas of deep learning**, such as memory bank in unsupervised and semi-supervised learning [1-2].
>
> A straightforward online scenario can be considered now: suppose we have a memory bank that stores the evaluated samples, and at regular intervals, we can add the online stream data to our memory bank. In this way, we can ensure that the sample size is large enough such that its marginal distribution can satisfy the ideal one. On the other hand, this time interval may be long enough so that the required computational overhead is minimal in comparison to the training-aware methods. Meanwhile, because our approach enjoys the strength of its convergence guarantee. In the end, we can extend OT to online scenarios.
>
> Your suggestion suddenly reminds us that we have carried out a similar experiment in the appendix. However, we forgot about this experiment during the subsequent discussion, for which we feel sorry that we could have used it earlier to illustrate the point that it is possible to fit our approach to the online scenario. So we are very grateful for the inspiration you have provided us with.
>
> In the appendix, we extend our OT to semi-supervised imbalanced learning. **The setup of this experiment can actually somewhat simulate the online scene**. Consider a semi-supervised training iteration in which a batch of data will contain both labeled and unlabeled samples. For the unlabelled data, we can access the pseudo predictions easily. However, because there are relatively little data in a batch, the OT cannot be applied to this batch. In this experiment, we did use a memory bank to store the historical predictions of each unlabeled sample, and at certain intervals, we rectified the predictions of the unlabeled samples in the memory bank with the OT and updated them on time. Experimental results also indicate that our method has a better performance compared to the state-of-the-art semi-supervised long-tail algorithm. Although this is not a strictly online situation, the experiment goes some way to showing that our approach can indeed be applied in an online situation.
>
> ### **Performance of the method under the estimated marginal distribution**
>
> This is a critical task. We are currently engaged in deliberating and implementing this setting, which we believe will considerably extend our existing algorithms. We will include this experiment in a future version as soon as possible.
>
> Lastly, thank you very much for your ongoing interest in our work, and your comments have made it much improved.
>
>
>
> [1] Unsupervised feature learning via non-parametric instance discrimination, CVPR 2018.
>
> [2] Semi-Supervised Deep Learning with Memory, ECCV 2018.

---

### Official Review · Reviewer_fcxC · 2021-11-04

**Correctness:** 3
**Technical Novelty And Significance:** 3
**Empirical Novelty And Significance:** 3
**Recommendation:** 6
**Confidence:** 4

**Main Review:**

Strengths
1. The idea to perform post-hoc correction from the perspective of optimal transport seems novel. Even though the detailed operation in practice is not complicated. Such an idea is well supported by the existing theory guarantees.

2. The results in the experiments also seem good. The proposed method can be applied to different baselines to boost their performance.

Weakness
1. Since this paper has many mathematical expressions, I suggest the authors give more detailed descriptions of the notations used. For example, in Section 2, what is r, c in U(r, c)? Is the c in equation (5) the same as c in U(r, c)?

2. For definition 1 and 2, the author may remark more for better understanding, especially when the notions used are not directly related to the proposed method. The introduction of some notations also seems missing or hard to get.

3. In equation (11), are the parameters W and Y optimized iteratively? The introduction of W also seems abruptly. A more natural and smooth transition between 3.1 and 3.2 may be needed.

4. To be honest, according to the description in the introduction, I did not get well about the difference between the proposed method and the previous Logit Adjustment. Maybe more details are needed.


**Summary Of The Paper:**

This paper proposes a new method for post-hoc correction in long-tailed recognition. Specifically, they leverage the idea of optimal transport (OT) and propose a linear mapping to replace the original exact cost matrix in OT problem. From the experiments, the proposed method can be combined with existing methods and boost their performance further.

**Summary Of The Review:**

Overall, the idea from OT to deal with long-tailed problems is novel. However, the writing of this paper makes it not very easy to understand or follow. Especially for people who don't have much background on OT. I thus give a score of 5.

================================update===================================

The author addresses most of my concerns. Also, the latest version has admitted its limitations and put a more clear clarification on its advantages. I thus decide to raise the score to boardline accept.

---

> ### Author Response · Authors · 2021-11-12
> **Response to reviewer fcxC**
>
> We thank the reviewer for your constructive comments. We are encouraged that you discovered that performing post-hoc correction from the perspective of optimal transport seems novel. We answer your questions below, and we will incorporate all feedback in the updated revision.
>
> ###  Q1: I suggest the authors give more detailed descriptions of the notations used.
>
> We have given more descriptions of the notations in the paper. For example, in Section 2, $U(\boldsymbol{r},\boldsymbol{c})$ include all matrices with row and column sums $\boldsymbol r$ and $\boldsymbol c$ respectively. To distinguish $\boldsymbol r$ in $U(\boldsymbol{r},\boldsymbol{c})$ from $\boldsymbol r$ in equation 5, we turn $\boldsymbol r$ in equation 5 into $\boldsymbol \mu$ , and $\boldsymbol \mu$ in equation 5 denotes the ideal distribution on the test set.
>
> ### Q2: For definition 1 and 2,  the author may remark more for better understanding
>
> We have reorganized and polished this section of OT to make it more accessible to those who are not familiar with the field; in general, we have started by defining the OT distance in the continuous case and then extended it to the OT in the discrete case. Finally, We present a lemma to illustrate the convergence of the  Sinkhorn algorithm.
>
> ###  Q3: are the parameters W and Y optimized iteratively
>
> Yes, $\boldsymbol W$ and $\boldsymbol Y$  are optimized iteratively until convergence.
>
> ### Q4: A more natural and smooth transition between 3.1 and 3.2 may be needed.
>
> We have smoothed the transition between section 3.1 and section 3.2. More specifically, we point out the limitations of the previous OT, simple functions is likely to be sub-optimal for the real data sets. This suggests the design of a better cost function to fit the long-tailed recognition problem better. However, manually designed cost functions require expert knowledge in different domains. Thus, we propose to use linear mapping to automatically learn the cost function, which relieves the need to configuration.
>
> ### Q5: Maybe more details about logit adjustment are needed
>
> In Preliminaries section, We have added a subsection to briefly explain the Logit Adjustment algorithm so that the reader can understand the advantages and disadvantages of this algorithm,  also better understand our contribution in this paper.

---

### Author Response · Authors · 2021-11-12
**Updated revision with new experiments, analyses, and clarifications**

1. We have reformulated and redefined the various notations and described them in more detail to reduce misunderstandings.
2. The OT section has been reorganized and embellished so that even those who are not knowledgeable in the field can quickly understand the section.
3. We have smoothed the transition between section 3.1 and 3.2 for better reading pleasure.
4. We have provided an introduction to Logit Adjustment so that readers can understand its advantages and disadvantages.
5. We have discussed when our approach is more applicable and when the sample-wise correction approach is more suitable.
6. We have highlighted more in the paper why we use linear mapping and the benefits of doing so.
7. We give some instructions on how to estimate the marginal distribution of the test data set in the case where it is unknown.
8. We have included a parametric study on $\lambda$ to understand its impact on the final performance in the appendix.

---

### Author Response · Authors · 2021-11-26
**To all reviewers**

Thanks to the reviewers for their great efforts and time. We thank all reviewers for their valuable comments. In the process of discussions with you, we have made our work more solid and thorough.

Thanks to reviewer vjwX and uP2E as they made us aware that our work is less practical with a smaller number of samples. Instead it is more feasible to apply our work in offline situations and also in the case of larger sample sizes. Thanks to reviewer fcxC  for illustrating our lack of mathematical notations and some relevant background knowledge, such as OT and Logit Adjustment.

We are very pleased and motivated that all reviewers agree that our approach of performing post-hoc correction from OT is very interesting and technically sound, with very good theoretical guarantees. Our results on the three datasets are also very substantial, especially on the iNaturalist dataset. As this dataset contains more than 8000 classes, our method is able to improve the accuracy by a further 3%. In addition, our methods can be freely cross-coupled with various training-aware methods.

We realize that it is almost impossible that the data set is perfectly balanced in practice.  Complying with some of the assumptions of the previous literature in this field, we realize that this assumption is a bit too strict. But there are still ways to widen our approach to more practical cases. **In the case of known marginal distributions of the dataset, we would like to clarify that our approach is not limited to the scenario of a balanced dataset**. **Specifically, in equation (10) in our latest revision**, **$\boldsymbol \mu$ can represent any desired target distribution, not just a balanced one.**  In this way, our approach is not merely restricted to the balanced case. And there is a fair amount of work [1-2] here that can be used to estimate the marginal distribution of the data set.

Last but not least, we would like to thank all reviewers again for their valuable comments and for their efforts. If there are any further suggestions or comments, please feel free to raise them and we will always be happy to answer your questions and resolve your concerns. In closing, thank you to all the reviewers.

[1] Using trusted data to train deep networks on labels corrupted by severe noise.

[2] Regularized learning for domain adaptation under label shifts.

---

### Public Comment · ~ZIYUN_LI1 · 2023-01-26
**Why not optimazing model m (instead of parameters W) and Y iteratively?**

Thanks for the instereting work, and I have a question below：

If I understand, the whole training processing is:
1. First train model m on imbalanced training set.
2. After achieving the fixed model m, optimize the parameters W (single layer) and Y iteratively to further improve performance.

Instead of optimizing the parameters W, how about the performance optimazing the model m and Y iteratively?

---

### Decision · Program_Chairs · 2022-01-20

**Decision:**

Accept (Poster)

**Comment:**

All reviewers agreed that the idea proposed by the paper is interesting and is well-motivated for handling long-tailed recognition problems.
As suggested by the reviewers, it seems important that the limitations the paper be addressed in the final version of the paper.